# Review of *Nephelium lappaceum* and *Nephelium ramboutan-ake*: A High Potential Supplement

**DOI:** 10.3390/molecules26227005

**Published:** 2021-11-19

**Authors:** Jia Ling Tsong, Lucky Poh Wah Goh, Jualang Azlan Gansau, Siew-Eng How

**Affiliations:** Faculty of Science and Natural Resources, Universiti Malaysia Sabah, UMS Road, Kota Kinabalu 88400, Sabah, Malaysia; bs17110261@student.ums.edu.my (J.L.T.); luckygoh@ums.edu.my (L.P.W.G.); azlanajg@ums.edu.my (J.A.G.)

**Keywords:** *Nephelium lappaceum*, *Nephelium ramboutan-ake*, antioxidant, bioactive, chemical

## Abstract

*Nephelium lappaceum* (*N. lappaceum*) and *Nephelium ramboutan-ake* (*N. ramboutan-ake*) are tropical fruits that gain popularity worldwide due to their tastiness. Currently, their potential to be used as pharmaceutical agents is underestimated. Chronic diseases such as cancer, diabetes and aging have high incidence rates in the modern world. Furthermore, pharmaceutical agents targeting pathogenic microorganisms have been hampered by the growing of antimicrobial resistance threats. The idea of food therapy leads to extensive nutraceuticals research on the potential of exotic fruits such as *N. lappaceum* and *N. ramboutan-ake* to act as supplements. Phytochemicals such as phenolic compounds that present in the fruit act as potent antioxidants that contribute to the protective effects against diseases induced by oxidative stress. Fruit residuals such as the peel and seeds hold greater nutraceutical potential than the edible part. This review highlights the antioxidant and biological activities (anti-neoplastic, anti-microbial, hypoglycemic actions and anti-aging), and chemical contents of different parts of *N. lappaceum* and *N. ramboutan-ake*. These fruits contain a diverse and important chemical profile that can alleviate or cure diseases.

## 1. Introduction

The demand for natural antioxidants is gaining attention because of their high therapeutic value. Antioxidants contribute greatly towards the prevention of disease development. Various studies have shown that antioxidants are able to prevent diseases such as atherosclerotic heart disease [1], neurodegenerative diseases [2], cancer [3], diabetes [4], and rheumatoid arthritis [5]. Tropical fruits are one of the promising antioxidant sources. High antioxidant activities were detected in tropical fruits such as guava [6], African mango pulp [7] and passion fruit [8]. Interestingly, antioxidant activities are not only detected in the edible part of the fruits, but also in the by-products such as the peel and seeds of the fruit [9,10]. Recently, the antioxidant properties of more exotic fruits have been reported [11]. However, studies on the potential of exotic fruits such as *Nephelium lappaceum* and *Nephelium ramboutan-ake* are lacking.

*N. lappaceum* and *N. ramboutan-ake* are tropical fruits in the *Sapindaceae* family. These trees are mostly located in tropical countries such as Malaysia, Thailand, Indonesia, Singapore, Sri Lanka and Philippines [12]. *N. lappaceum* has also been cultivated in the warmer region of south China, Taiwan, Australia and Hawaii [13]. *N. lappaceum* is known as rambutan, and *N. ramboutan-ake* are known as pulasan by the locals. “Rambutan” is a word originating from the Malay “rambut”, which means “hair”, whereas “pulasan” is a word originating from the Malay “pulas”, means twist [12].

Both, *N. lappaceum* and *N. ramboutan-ake* can be eaten freshly or be further processed into pickles or jams. The seeds can be roasted and eaten. Apart from that, they also act as traditional medicines. The root decoction of *N. lappaceum* and *N. ramboutan-ake* can be used to treat fever. In addition, the leaves and roots can be applied as poultices to relieve headache [14]. Other functions of *N. lappaceum* which have been described by locals include vermicide, diarrhoea treatment, and dysentery treatment, while a bark decoction can be used to treat thrush [14]. According to the native people, *N. ramboutan-ake* is also used as a traditional treatment for scabies and itchiness [15].

Local traditional treatments and food therapy have led to an increased interest in their potential therapeutic effects. The therapeutic effects and bioactivities of fruits and vegetables are mainly contributed by phytochemicals which come from phenolic compounds [16]. Most bioactive components are from the flavonoid groups such as anthocyanin, flavonols, flavanones, flavones and flavans. However, the phenolic contents of exotic fruits such as *N. lappaceum* and *N. ramboutan-ake* remain unexplored, and should thus receive attention from nutraceutical scientists to discover more potent drug candidates for disease treatments. The bioactive components of fruits and their relative bioactivities serve as a new hope and alternative option to solve the current disease dilemma faced by the medical field. Antioxidants present in fruits have the capability of reducing the risk of developing chronic diseases such as cancer, pathogenic disease, diabetic and aging [1,3,4].

This article reviews the antioxidant activities, bioactivities and chemical profile in different parts of *N. lappaceum* and *N. ramboutan-ake*. It provides new insight for researchers to further study the potential of *N. lappaceum* and *N. ramboutan-ake* to act as nutraceuticals.

## 2. Chemical Profile of *N. lappaceum*

### 2.1. Peel

The major chemical compounds that are present in the peel of *N. lappaceum* are ellagitannins, gallotannins, hydroxycinnamic acids, hydroxybenzoic acids and flavonoids (Table 1). Geraniin, corilagin, and ellagic acid are the most abundant bioactive compounds in the peel [17,18,19,20]. Similar results were also reported by other researchers, who identified ellagic acid, geraniin and galloylshikimic acid as the major phenolic compounds in the peel of *N. lappaceum* [21]. On the contrary, another study described that p-coumaric acid and rutin were the main compounds in *N. lappaceum* peel [22]. Hydroxycinnamic acids such as caffeic acid were found in *N. lappaceum* [22]. Specifically, Phuong et al., 2020 reported that the major component in *N. lappaceum* peel was geraniin (397.28 mg/g), followed by ellagic acid (176.99 mg/g), quercetin (167.37 mg/g), rutin (36.40 mg/g) and corilagin (3.81 mg/g) [23]. Anthocyanins such as pelargonidin and Vitisin A were detected in *N. lappaceum* peel [18]. Pro-vitamin A was found to be significantly higher in peel than other parts, as it is responsible for the peel colour [24]. Small amounts (0.02 to 0.31 mg/100 mL) of vitamins such as thiamine, riboflavin, and niacin were also detected in dry and fresh peel. Interestingly, the carotene level in peel increases drastically from 10.60 to 41.20 µg/100 g when the peel is dried compared with the undried peel [25]. Given the wide and diverse phytochemical contents of *N. lappaceum* peel, there is great potential in developing supplements from its peel.

The oral toxicity of geraniin- and geraniin-enriched extracts revealed that the LD_50_ cut off value of geraniin- and geraniin-enriched extracts (2000 mg/kg body weight) had no significant adverse effects to rat’s body weight, water intake, histopathological, haematological and biochemical metabolites. However, a study found that rats experienced diarrhoea and the histopathology stain of the hepatocyte was observed to be ‘foamy’ [26]. Hence, caution should be taken while exploring the therapeutic effect of geraniin in *N. lappaceum* because of its potential to cause liver abnormalities.

### 2.2. Seed

The seeds of *N. lappaceum* are made up of crude fat (37.1–38.9%), protein (11.9–14.1%) and fibre (2.8–6.6%), with a moisture content of 34.1–34.6% and an ash content of 2.6–2.9% [27]. However, another study reported that the major constituent of the seeds is carbohydrate (48.1%), followed by fat (38.9%), protein (12.4%), moisture (3.31%) and ash (2.26%), with saponification (157.07%), iodine (37.64%) and free fatty acid (0.37%) contents [28].

The major components and composition of the fat have been further analysed by many researchers with different proportions of sub-classes reported. The major fatty acids detected are oleic acid and arachidic acid (Table 2). The percentages reported for oleic acids and arachidic acid were 31–42% and 28–37%, respectively. Other minor fatty acids such as stearic, palmitic, galodeic and linoleic acid were also present [28,29,30,31]. The presence of anti-nutrients such as saponin, tannin, phytate and oxalate was also determined in tolerable amounts [32,33]. The high amount of arachidic acid causes the seed fat in semi-solid form at room temperature with a high melting point which is suitable for use in the confectionary and cosmetic industries [29,30].

A study reported that a higher total phenolic content (TPC) was detected in peel, while a higher total flavonoid content (TFC) was determined in seeds [34]. The major phenolic compounds present in the seeds are flavonoids [19]. Flavonoid compounds such as kaempferol type compounds were detected in the seed extract (Table 2). The bioactivities exhibited by the seed extract includes anti-microbial effects [35], hypoglycaemic effects [36], and anti-aging [19]. These bioactivities have been shown to be contributed by flavonol glycoside and acylated flavonol glycosides in the seeds [19].

### 2.3. Pulp

The pulp of *N. lappaceum* is reported to contain sugar, organic acid and vitamin (Table 3). The sugar content is mainly composed of sucrose (5.38–10.01%), followed by fructose (1.75–3.18%) and glucose (1.72–2.43%). Organic acids such as citric acid (35.8–74.3%) is the major compound, found followed by lactic acid, ascorbic acid, malic acid and tartaric acid [33].

Various vitamins are also present in the pulp, including ascorbic acid, niacin, riboflavin and thiamine. Ascorbic acid (58.29 mg/100 g) is found to be more abundant in the pulp as compared with the peel and seeds. Niacin, riboflavin and thiamine are present in low concentrations in fresh pulp (0.02 to 0.78 mg/100 g) and dried pulp (0.01 to 0.56 mg/100 g) [25]. However, a lower value of ascorbic acid (60.89 μg/g or 6.09 mg/100 g) in the pulp was reported by another study [24]. This may be due to the different cultivars and the extraction methods used in the two studies. *N. lappaceum* pulp also contains a wide variety of phytochemical compounds that are responsible for the odour, such as β-damascenone, ethyl 2-methylbutyrate, 2,6-nonadienal, (E)-2nonenal, and nonanal which are the main contributors to fruit aroma [37].

## 3. Antioxidant Activities of *N. lappaceum*

Different parts of the *N. lappaceum* have been subjected to in vitro antioxidants activities evaluation (Table 4). The peel of *N. lappaceum* is found to exert higher antioxidant activities as compared with the seed and leaf [21,34,38,39]. Higher antioxidant activities in the peel are attributed to higher phenolic and flavonoid contents [32,40]. The wide varieties of phenolic compounds such as ellagitannin, phenolic acid, quercetin and gallotannin in the peel have been associated with higher antioxidant activities. In addition, although vitamins in the peel are present in small quantities, they also act as antioxidants [25].

A study reported that the peel consists of high TPC, whereas the seed extract contains higher TFC than the peel [34]. This can be explained by the presence of kaempferol compounds in the seeds [19]. Although there are antioxidants activities reported in the pulp, it shows relatively low antioxidant properties, which is mainly contributed by ascorbic acid [24,41,42]. Hence, the antioxidant activities and capabilities of different part of fruit are dependent on their phytochemical composition. There is a wide range of antioxidants present in peel including phenolic acid, ellagitannin, flavonoid and vitamin. For seeds, the antioxidant activities are contributed by flavonoids, whereas for pulp, the antioxidant activities depend on vitamins.

Geraniin is a potent and major antioxidant found in *N. lappaceum*. It shows high antioxidant properties among the tested bioactive compounds [17]. The antioxidant properties of the peel are able to reduce the complications caused by oxidative stress. An in vivo experiment observed that rats fed orally with the peel extract showed a reduction in lipid accumulation in the liver and reduction in malondialdehyde (MDA) by inhibiting the expression of peroxisome proliferator activating receptor γ (PPARγ). This indicates that the peel has the potential to prevent liver damage from oxidative stress [43,44]. Apart from that, a reduction in the total leucocytes and protective effect on lungs has been shown in rats treated with cigarettes and peel extract together [45]. A study reported that peel extract was able to boost the antioxidant activities of mice. The positive signs of enhanced antioxidant activity include a reduction in reactive oxygen species (ROS) formation and a reduction in human liver cancer cell (HepG-2) cell apoptosis induced by H_2_O_2_. D-galactose-induced aging mice showed an increase in total antioxidant capacity (T-AOC) when treated with peel extract. Moreover, positive signs of antioxidant defence systems such as a reduction in MDA and an increase in superoxide dismutase (SOD) and glutathione peroxidase activity (GSH-Px) were demonstrated in different mice tissues [44]. These results were supported when photo-aging mice treated with peel extract displayed similar antioxidant defence system [46].

Phenolic compounds correspond to the antioxidant properties exerted by fruits. Therefore, the higher the phenolic content in a fruit, the greater the antioxidant activity exerted. Strong positive correlations are reported between TPC and antioxidant activity in *N. lappaceum* [21,39]. The peel extract with higher TPC shows higher DPPH radical scavenging effect than the seeds with higher TFC [34].

The solvent used during antioxidant extraction is found to have a significant effect on antioxidant activities. Higher antioxidant activity is detected in ethanolic extract of *N. lappaceum* as compared with aqueous extract [38]. Furthermore, ethanol extract from the fruit contains higher TPC than the aqueous and methanol extract [22]. A similar result was observed when ethanol extract showed higher TPC than aqueous extract for peel (244 mg GAE/g) and seeds (27.1 mg GAE/g) (Yunusa et al., 2018). In addition, ether, methanolic and aqueous extract of *N. lappaceum* were analysed for TPC and antioxidant activities [39]. As a result, the highest TPC (542.2 mg GAE/g) and antioxidants activities (IC_50_ = 4.94 μg/mL) were shown by the methanol extract as compared with other solvents (Table 4). The better extraction property of methanol was further supported by another study where methanol extract exhibited better anti-neoplastic effects as compared with other solvent extractions [47]. However, another study indicated that the use of ethyl acetate contributes to high antioxidant activities [48]. This is due to the difference in solubility of active compounds in solvents with different polarities [26].

Other extraction variables such as technology, liquid–solid ratio, extraction time and temperature may affect TPC extraction. By using microwave-assisted extraction with 80.85% ethanol solvent, the resulting extract contains the highest TPC (213.76 mg GAE/g) [22]. This is supported by another study which observed 80% of ethanol extract yielded the highest TPC (397.06 mg GAE/g) [49]. In addition, the fruit peel extracted with ultrasound-assisted 10% ethanol in a mass volume of 1/7 had the highest TPC (487.67 mg GAE/g) [20] (Table 4). It can be concluded that alcoholic solvent is the best option for antioxidants extraction, as phenolic compounds dissolve more readily in an alcohol solvent compared with other solvents.

Moreover, antioxidant activities and capabilities can be affected by the cultivar of *N. lappaceum* and the growing environment of the tree. Cultivar Sichompu shows higher inhibition of ROS formation as compared with cultivar Rongrien [42]. Cultivar Binjai poses higher ABTS antiradical activity while cultivar Aceh demonstrates higher activity in FRAP assay [48]. The bitter variety of *N. lappaceum* exhibits higher activity in all antioxidant assays tested when compared with the sweet varieties [40]. This is because the phytochemical compositions are different in cultivars caused by environmental factors such as temperature, humidity, pH and nutrient in the soil that affect the metabolism of the plant and thus affects the secondary metabolite production.

## 4. Other Biological Activities of *N. lappaceum*

### 4.1. Anti-Neoplastic Effects

The anti-proliferative effect of *N. lappaceum* peel extract has been tested on a cervical cancer cell line (HeLa), breast cancer cell line (MDA-MB-231) and osteosarcoma cell line (MG-63) [50]. The study observed that both the red and yellow peels of *N. lappaceum* exhibited anti-proliferative effects on breast cancer cell lines and osteosarcoma cancer cell lines. The yellow peel extract of *N. lappaceum* showed a better anti-proliferative effect on the breast cancer cell line (IC_50_ = 5.42 μg/mL) and osteosarcoma cell line (IC_50_ = 6.97 μg/mL) as compared with the red peel extract.

A recent study revealed that the anti-neoplastic effects of methanol peel extract successfully reduced HepG-2 cancer cell line viability via inducing apoptosis with DNA fragmentation and cell shrinkage. Phytochemical analysis showed that the high anti-neoplastic effects of methanol peel extract were contributed by the high phenolic contents such as coumarin, flavonoid, phenols, saponin and tannin. Other compounds, such as alkaloid, carbohydrate, cardiac glycoside, protein, terpenoid and triterpenoid, were also detected in the methanol extract [47]. However, the anti-proliferative effect of peel and seed of the fruit tested on human mouth carcinoma cell line (CLS-354) showed that the peel (IC_50_ = 292 μg/mL) and seeds (IC_50_ = 305 μg/mL) did not show high anti-proliferative effects on the cancer cell lines [9].

A trypsin inhibitor extracted from the seed was found to exhibit a dose-dependent anti-tumour effect on breast cancer cell (MCF-7), HepG-2 and nasopharyngeal carcinoma cell lines (CNE-1 and CNE-2) [51]. In addition, a 22.5-kDa trypsin inhibitor from *N. lappaceum* seeds has an inhibitory effect on HIV-1-reverse transcriptase activity and reduces the proteolytic activities of trypsin as well as α-chymotrypsin through disulphide bonds. The isolated trypsin inhibitor is one of the few inhibitors that can stimulate the production of nitric oxide, which acts as an anti-tumour molecule [51].

### 4.2. Anti-Microbial

The anti-microbial activities of the fruit against different microorganisms are shown in Table 5. The growth of *Staphylococcus aureus* is inhibited by peel extract [22,39,52,53,54]. In addition, the growth of multi-drugs resistant *Staphylococcus aureus* (MRSA) is also inhibited when treated with methanol peel extract (MIC = 0.4 mg/mL) [53] and ethanol peel extract (MIC= 0.98–1.95 mg/mL) [55]. Apart from that, the peel extract also shows inhibitory activity against other microorganisms such as yeast *Candida* sp. [23,53], whereas the leaf extract exhibits anti-microbial activity against multi-drug resistant *Pseudomonas aeruginosa* [56].

Most of the studies using methanol peel extracts did not observe inhibitory effects against *Escherichia coli* (*E. coli*) [39,53,54]. However, inhibitory effect was detected in *E. coli* when treated with ethanolic peel extract with a zone of inhibition of 15 mm [53]. Nevertheless, there is a study that observes the great potential of methanol peel extract against a wide variety of Gram-negative bacteria, including *E. coli* and *Pseudomonas aeruginosa* [23].

Anti-bacterial activities are observed to be correlated to the phenolic content. Methanol extract of the peel consists of higher TPC and thus exert better inhibitory effect as compared with seed extract [39]. Furthermore, bioactive components such as flavonoid, tannin and triterpenoid are present in the peel but not the seed. Bioactive compounds such as geraniin, ellagic acid, rutin, quercetin, and corilagin in the peel of *N. lappaceum* most likely contribute to the antimicrobial properties [23].

Nonetheless, some anti-microbial activity could be detected in the seed extract [35]. The aqueous seed extract with protein content shows anti-bacteria activity on *Streptococcus pyogenes*, *Bacillus subtilis*, and *S. aureus*. Similar to the peel extract, the most sensitive bacterial strain when treated with seed extract is *S. aureus*. Moreover, there is a weak anti-bacterial activity against *E. coli* (6.5 mm) and *P. aeruginosa* (10 mm). This shows that the anti-bacterial activity of the seeds is not solely dependent on the phenolic content but also the protein content.

### 4.3. Hypoglycemic Actions

Geraniin extracted from the peel of *N. lappaceum* was found to exhibit a hypoglycemic effect on carbohydrate enzymes such as α-glucosidase and α-amylase in addition to antioxidant activities [58]. Apart from that, geraniin shows higher inhibition of aldose reductase and age activities when compared with a control treatment. These two activities are the culprits for complications of diabetic. Hence, the peel of *N. Lappaceum* can be further developed as a supplement for diabetic patients. The bioactive compounds other than geraniin in the peel also carry hypoglycemic actions. The peel extract shows better inhibition activities of α-glucosidase (ic_50_ = 0.106 µg/mL) and β-glucosidase (ic_50_ = 7.02 µg/mL) than geraniin. A-glucosidase breaks down starch to glucose, which contributes to high glucose levels. Thus, the inhibition of α-glucosidase is one of the approaches to induce hypoglycemic. The hypoglycemic actions are reported to have strong positive correlation with radical scavenging activity of the peel extract [59].

The hypoglycemic actions of *N. lappaceum* were studied in vivo, in which the reduction in α-glucosidase and α-amylase enzyme activities were observed [60,61]. Moreover, rats treated with *N. lappaceum* extract showed signs of anti-hyperglycemia such as reduction in blood glucose level and improvement of glucose tolerance. Organ damage such as to the liver, pancreas and kidney was not observed in diabetic rats treated with *N. lappaceum* peel extract [60,61]. This result is supported by a study where rats that were orally administered geraniin from the peel did not show significant adverse results in body weight and organ, even though higher feed was taken by the rat [62].

### 4.4. Anti-Aging

Anti-aging properties have been discovered in the peel as well the pulp of *N. lappaceum* [63]. A study suggests that the peel and pulp extracts of the fruit are potent antioxidant agents that can be formulated into skin care products for preventing skin aging through the tyrosinase inhibition activities of peel (IC_50_ = 38.88 μg/mL) and pulp (IC_50_ = 43.80 μg/mL) extracts [64].

A pre-treatment with peel extract on HepG-2 cell lines observed a reduction in apoptosis. Clear attenuations of liver and kidney damage were also observed in aging mice treated with the peel extract [44]. The synergistic effect of the peel extract and Leu-Ser-Gly-Tyr-Gly-Pro (LSGYGP) on anti-aging reveals that the combination of peel extract and LSGYGP lead to an increase in hyaluronic acid and hydroxyproline, a decline in proinflammatory cytokines, collagenolytic enzyme (matrix metalloproteinase) and reduction in the mitogen-activated protein kinases (MAPK) signalling pathway, which contribute to the anti-aging effect [46].

The seeds of the fruit also contribute to the anti-aging effect by means of other mechanisms [19]. The inhibition effect of cellular aging was driven by factors such as SA-β-gal activity, CDKIs and p16INK4A. On the other hand, a significant increase in SIRTs level is manifested by three flavonoid compounds (kaempferol 3-O-β-d-glucopyranosyl-7-O-α–l-rhamnopyranoside, kaempferol 3-O-rutinoside and kaempferol 7-O-α-l-rhamnopyranoside) isolated from the seed extract [19]. Therefore, flavonoid compounds from *N. lappaceum* play a significant role in anti-aging activity.

## 5. Chemical Profile of *N. ramboutan-ake*

Fruit rinds of *N. ramboutan-ake* have been observed to contain phytochemicals [65], such as the presence of steroid in hexane extract; alkaloid and terpenoid in ethyl acetate extract; alkaloid, phenolic, flavonoid and terpenoid in methanol extract. It is also observed that more phytochemicals can be extracted with alcoholic solvent when compared with non-alcoholic solvent. However, studies on bioactive compounds from the peel, seeds and pulp of the fruits of *N. ramboutan-ake* still remain scarce. 

## 6. Antioxidant Activities of *N. ramboutan-ake*

Currently, antioxidant activities of *N. ramboutan-ake* have only been tested in vitro (Table 6). Most of the antioxidant activity assessments were performed using only the peel [65,66,67]. This may be due to the research based on *N. lappaceum* peel suggesting that it has a better potential for further development as compared with the less well-studied *N. ramboutan-ake*. The other parts of *N. ramboutan-ake*, including the leaves [68] and pulp [24,69], are also reported. Antioxidant assays that are reported mostly on *N. ramboutan-ake* are DPPH radical scavenging assay and FRAP. Other assays such as the beta-carotene bleaching assay, the galvinoxyl assay and lipid peroxidation are also reported. Furthermore, various antioxidant activities have been detected in the fruit [66] (Table 6).

DPPH radical scavenging activities of the methanolic peel extract were reported to have an IC_50_ of 0.019 mg/mL [66] and an IC_50_ of 57.389 mg/mL, which varied quire significantly, possibly due to different cultivars [65]. The antioxidant activity of leaf was also reported with an IC_50_ of 0.244 mg/mL [68]. Most of the extraction solvents used were methanol [65,66,69] and ethanol [67,68] (Table 6). This is because extracts obtained using alcoholic solvents show better antioxidant activity [67,68], and more phytochemicals such as alkaloid, terpenoid, phenolic and flavonoid compounds are extracted [65].

A comparative study conducted on different fruit juices including *N. ramboutan-ake* and *N. lappaceum* observed that antioxidant activities such as DPPH and FRAP of the *N. ramboutan-ake* and *N. lappaceum* are lower compared with other fruit juices [24]. This can be explained by the relatively low TPC of *N. ramboutan-ake* (199.07 μg GAE/g) as compared with *N. lappaceum* (TPC = 223.75 μg GAE/g). The antioxidants of the pulp may be attributed to vitamins, though fewer phenolic compounds are present in the pulp of *N. lappaceum*. Interestingly, *N. ramboutan-ake* shows a strong metal chelating effect of 72.78%.

The relationship between TPC and antioxidant properties of *N. ramboutan-ake* is a positive correlation with R^2^ of 0.738 between TPC and DPPH radical scavenging activity [68]. These findings are supported by another study on the same fruit where a positive correlation with R^2^ of 0.655 and 0.828 between TPC and antioxidant activities for DPPH and FRAP, respectively, is observed [67].

However, a study conducted on different types of underutilised fruits pulp including *N. ramboutan-ake* showed a contrasting result with a weak negative correlation with R^2^ of −0.062 [69]. This indicates that the pulp may exhibit relatively low TPC and antioxidants when compared with the peel. The list of antioxidant assays performed on different parts of the fruit is shown in Table 6. To date, no antioxidant assay has yet been performed on the seeds of *N. ramboutan-ake*.

## 7. Other Biological Activities of *N. ramboutan-ake*

### 7.1. Anti-Neoplastic Effects

The potential of *N. ramboutan-ake* on anti-neoplastic effects is observed where the aqueous fraction of the rind is able to reduce the cell viability of HT-29 (human colorectal adenocarcinoma cell, HTB-38), HCT-116 (human colorectal carcinoma cell, CCL-247), Ca Ski (human caucasoid cervical carcinoma cells, CRL1150) and MDA-MB-231 (human breast adenocarcinoma cells, HTB-26) cell lines in a dose-dependent manner [70].

Among the cancer cell lines, HT-29 shows a significant decline in cell viability. The rinds contain various phytochemicals that have been demonstrated to induce apoptosis in cancer cell lines. The HT-29 apoptosis induced by *N. ramboutan-ake* extract involves the dysfunction of mitochondria which is confirmed by the elevated level of Bax protein and caspases. Dysfunction of mitochondria will trigger the cellular apoptosis cell signalling pathways, such as an increased ROS level and a reduction in the RSH level, which triggers apoptosis [70].

Moreover, the cytotoxicity level of *N. ramboutan-ake* extract had been investigated by measuring the cell proliferation effect of ethanolic extract of *N. ramboutan-ake* on 3T3 (mouse embryonic fibroblasts cell) and 4T1 (mouse breast cancer cell) cell lines. The findings show that *N. ramboutan-ake* exhibits anti-proliferation toward both cell lines, which indicates that the fruit does not exhibit cytotoxic effect on the cell lines [68]. Hence, the peel of *N. ramboutan-ake* possesses important bioactive components that could be further developed as nutraceuticals for the prevention of colon cancer.

### 7.2. Anti-Microbial

Due to the inappropriate overuse and the extensive agriculture use of antibiotics, the rise of antibiotic resistance in bacteria has been occurring at an alarming level [71]. Therefore, scientists are paying great attention to phytochemicals in plant extracts in order to discover novel anti-bacterial compounds. The ethyl acetate and ethanol extract of *N. ramboutan-ake* peels and seeds are effective to suppress the growth of *S. aureus* and *E. coli*. The peel exhibits a better suppression effect on both bacterial strains as compared with the seed extract [72].

### 7.3. Hypoglycemic Actions

The non-edible part of the *N. ramboutan-ake* has also become a target in the search for potential bioactivities. The seed of *N. ramboutan-ake* has the potential of preventing diabetes and obesity [73]. A study shows that the peptide produced by *N. ramboutan-ake* seed protein can bind to the active site of α-amylase, which reduces the binding site of this enzyme and inhibits the hydrolysis of starch to glucose. The most effective gastro-digestive enzyme for protein in *N. ramboutan-ake* seed is pepsin [73]. The protein in *N. ramboutan-ake* seeds is more susceptible to multiple gastro-digestive enzymes, and therefore has higher inhibitory activity than *N. lappaceum* seeds.

## 8. A Comparison of Studies Conducted on *N. lappaceum* and *N. ramboutan-ake*

The present review further summarises the studies conducted on *N. lappaceum* and *N. ramboutan-ake*. The comparison in Table 7 illustrates that studies conducted on *N. ramboutan-ake* are lagging behind *N. lappaceum* in many aspects, including anti-aging, anti-neoplastic, antimicrobial and antioxidants. Studies conducted on *N. lappaceum* have demonstrated its tremendous potential to be further developed into nutraceuticals. However, the beneficial properties of *N. ramboutan-ake* could be further explored to reveal its true potential in future studies.

## 9. Conclusions

Antioxidant activities and capabilities vary in each part of the fruits. This is because of the difference in phytochemical composition. Generally, the peel and seeds exhibit higher antioxidants activities and bioactivities as there are more bioactive compounds present. Phenolic and flavonoid compounds are responsible for the antioxidant activities and other biological activities. In vitro and in vivo studies indicated that *N. lappaceum* carries pharmaceutical potential such as alleviating oxidative-stress related diseases. However, continuous effort should be carried out in future to determine the exact related bioactive components towards the development of nutraceuticals.

Unfortunately, the investigation into the phytochemicals and the in vitro and in vivo biological activities of *N. ramboutan-ake* is still lacking, which results in the underutilization of this fruit. Thus, the bioactive components responsible for the biological activities still remained unknown. Research on *N. lappaceum* can serve as a reference for enhancing knowledge in *N. ramboutan-ake*. Furthermore, limited evaluation of in vivo bioactivity remains a major challenge to study the pharmaceutical effects of both fruits. Toxicology and pharmacology of the bioactive compounds from fruits should be studied so that their therapeutic effect towards disease prevention can further be supported. In conclusion, this review has highlighted the potential and importance of *N. lappaceum* and *N. ramboutan-ake*. Hence, further research should be carried out in order to volarize the potential of both fruits, which can be used as nutraceuticals or food supplements.

## Figures and Tables

**Table 1 molecules-26-07005-t001:** Bioactive compounds from *N. lappaceum* peel.

Compound	Group	References
Geraniin	Ellagitannin	Thitilertdecha et al. (2010) [17] Hernandez et al. (2017) [18] Lee et al. (2020) [19] Mendez-Flores et al. (2018) [20] Nguyen et al. (2019) [21] Phuong et al. (2020) [23]
Corilagin	Ellagitannin	Thitilertdecha et al. (2010) [17] Hernandez et al. (2017) [18] Lee et al. (2020) [19] Mendez-Flores et al. (2018) [20] Phuong et al. (2020) [23]
Ellagic acid	Ellagitannin	Thitilertdecha et al. (2010) [17] Hernandez et al. (2017) [18] Mendez-Flores et al. (2018) [20] Nguyen et al. (2019) [21] Phuong et al. (2020) [23]
p-coumaric	Hydroxycinnamic acid	Sun et al. (2012) [22]
Caffeic acid	Hydroxycinnamic acid	Sun et al. (2012) [22]
Syringic acid	Hydroxybenzoic acid	Sun et al. (2012) [22]
Gallic acid	Hydroxybenzoic acid	Lee et al. (2020) [19] Nguyen et al. (2019) [21] Sun et al. (2012) [22]
Quercetin	Flavonoid	Phuong et al. (2020) [23]
Rutin	Flavonoid	Sun et al. (2012) [22] Phuong et al. (2020) [23]

**Table 2 molecules-26-07005-t002:** Chemical compounds from *N. lappaceum* seeds.

Compound	Group	References
Octadic-9-enoic acid	Fatty acid	Harahap et al. (2012) [28] Manaf et al. (2013) [29] Lourith et al. (2016) [30] Ghobakhlou et al. (2019) [31]
icosanoic acid	Fatty acid	Harahap et al. (2012) [28] Manaf et al. (2013) [29] Lourith et al. (2016) [30] Ghobakhlou et al. (2019) [31]
Kaempferol 3-O-β-d-galactopyranosyl-7-O-α–l-rhamnopyranoside	Flavonoid	Lee et al. (2020) [19]
Kaempferol 3-O-β-d-glucopyranosyl-7-O-α–l-rhamnopyranoside	Flavonoid	
Kaempferol 3-O-α-l-arabinopyranosyl-7-O-α–l-rhamnopyranoside	Flavonoid	
Kaempferol 3-O-rutinoside	Flavonoid	
Astragalin	Flavonoid	
Kaempferol 7-O-α-l-rhamnopyranoside	Flavonoid	

**Table 3 molecules-26-07005-t003:** Chemical compounds from *N. lappaceum* pulp.

Compound	Group	References
Sucrose	Disaccharide	Chai et al. (2018) [33]
Fructose	Monosaccharide	
Glucose		
Citric acid	Organic acid	
Lactic acid		
Ascorbic acid	Vitamin	Johnson et al. (2013) [25] Chai et al. (2018) [33]
Niacin		Johnson et al. (2013) [25]
Riboflavin		
Thiamine		

**Table 4 molecules-26-07005-t004:** List of antioxidant assays performed on *N. lappaceum*.

Part	Solvent	TPC	Antioxidant Assay	ABTS * (IC_50_)	FRAP *	DPPH * (IC_50_)	% of Activity	References
Pulp	Ethanol	-	ABTSDPPH	-	-	-	70	Leong and Shui (2002) [41]
Peel Leaf Pulp Seed Peel Leaf Pulp Seed	Aqueous Ethanol	300.0 ^a^ 108.0 ^a^ - - 762.0 ^a^ 390.0 ^a^ - -	DPPH ABTS Galvinoxyl Superoxide anion Lipid autooxidation Pro-oxidant assay	16.5 ^f^ 24.5 ^f^ - - 1.7 ^f^ 12.2 ^f^ - -	-	18 ^f^ 33 ^f^ - - 3.7 ^f^ 16 ^f^ - -	47.5 28.2 - - 41.4 6.6 - -	Palanisamy et al. (2008) [38]
Peel Seed	Ether Methanol Aqueous Ether Methanol Aqueous	293.3 ^b^ 542.2 ^b^ 393.2 ^b^ 7.4 ^b^ 58.5 ^b^ 3.3 ^b^	FRAP β-carotene Linoleic peroxidation DPPH	-	-	17.3 ^f^ 4.94 ^f^ 9.67 ^f^ - - -	-	Thitilertdecha et al. (2008) [39]
Peel	Ethanol	213.76 ^a^	FRAP DPPH OH scavenging Lipid Peroxidation Nitrite scavenging	-	-	3.55 ^f^	-	Sun et al. (2012) [22]
Peel Seed	Methanol	104.6 ^a^ 124.14 ^a^	DPPH ABTS	-		-	-	Chunglok et al. (2014) [9]
Pulp	Aqueous	223.75 ^c^	DPPH FRAP Metal Chelating	-	96.85 ^h^	-	3.39	Sulaiman and Ooi (2014) [24]
Pulp	Ethanol	-	H_2_O_2_ scavenging assay	-	-	-	25	Chingsuwanrote et al. (2016) [42]
Peel	Aqueous	457.0 ^a^	ABTS FRAP	38.24 ^f^	0.203 ^i^	-	-	Hernandez et al. (2017) [18]
Peel	Ethanol	487.67 ^a^	ABTS DPPH Lipid oxidation inhibition	-	-	-	92.50 73.73 91.74	Mendez-Flores et al. (2018) [20]
Peel	Methanol Petroleum ether Chloroform Ethyl acetate Aqueous	-	ABTS FRAP	0.76 ^f^ 6.98 ^f^ 0.76 ^f^ 0.77 ^f^ 0.52 ^f^ (Binjai)	864.53 ^j^ 132.29 ^j^ 883.76 ^j^ 1424.9 ^j^ 328.31 ^j^ (Aceh)	-	-	Mistriyani et al. (2018) [48]
Peel Seed	Ethanol Aqueous Ethanol Aqueous	244.00 ^a^ 49.92 ^a^ 27.1 ^a^ 7.93 ^a^	DPPH	-	-	24.99 ^f^ 144.59 ^f^ - -	95.73 80.25 1.67 1.90	Yunusa et al. (2018) [34]
Peel Seed	Methanol Methanol	12.68 ^d^ 0.12 ^d^	ABTS FRAP DPPH	54.09 ^g^ 0.32 ^g^	66.05 ^k^ 0.39 ^k^	46.38 ^g^ 0.11 ^g^	- -	Nguyen et al. (2019) [21]
Peel	Hydroalcoholic	23.98 ^e^	ABTS FRAP DPPH	651.70 ^f^	1407.81 ^l^	9.72 ^f^	94.22 90.82	Lopez et al. (2020) [20]
Peel	Acidic 1% Alkaline 1% Aqueous Ethanol Hydroethanolic 60%	231 ^a^ 262 ^a^ 280 ^a^ 208 ^a^ 340 ^a^	ABTS DPPH	-	-	-	-	Monrroy et al. (2020) [26]
Peel	80% Ethanol	397.06 ^a^	DPPH Nitric oxide scavenging β-carotene	-	-	8.87 ^f^	- 64.88 98.19	La et al. (2013) [49]

^a^ In mg of gallic acid equivalents (GAE) per g; ^b^ in mg of catechin equivalents per g; ^c^ in µg of gallic acid equivalents (GAE) per g; ^d^ in g of gallic acid equivalents (GAE) per hundred g; ^e^ in mg per mL; ^f^ in µg per mL; ^g^ in g of trolox equivalents (TE) per hundred-g decimeter; ^h^ in µg of trolox equivalent antioxidant capacity (TEAC) per g; ^i^ in gallic acid equivalents (GAE) per mL; ^j^ in µg per mg; ^k^ in g ferrous (II) ion equivalent per hundred-gram decimeter; ^l^ in µmole per L. * Abbreviations: ABTS: 2,2′-azino-bis(3-ethylbenzothiazoline-6-sulfonic acid) diammonium salt; DPPH: 2,2-diphenyl-1-picrylhydrazyl; FRAP: ferric reducing antioxidant power; TPC: total phenolic compound.

**Table 5 molecules-26-07005-t005:** List of anti-microbial tests performed on *N. lappaceum*.

Part	Antimicrobial Activities	MIC (mg/mL)	Zone of Inhibition (mm)	References
Peel	*Staphylococcus aureus* *Bacillus cereus* *Proteus vulgarius* *Salmonella typhi* *Bacillus subtilis* *Escherichia coli* *Candida lipolytica*	0.5 1.0 1.0 0.5 1.0 1.0 0.5	27 14 15 26 15 15 30	Mohamed et al. (1994) [53]
Peel	*Pseudomonas aeruginosa* *Vibrio cholera* *Enterococcus faecalis* *Staphylococcus aureus* *Staphylococcus epidermidis*	62.5 15.6 15.6 31.2 2.0	7.75 17.25 11.75 8.50 16.50	Thitilertdecha et al. (2008) [39]
Peel	*Staphylococcus aureus* *Methicillin-resistant S. aureus* (MRSA) *Streptococcus mutans*	- - -	20.2 19.2 8.5	Tadtong et al. (2011) [53]
Peel	*Streptococcus pyogenes* *Staphylococcus aureus*	- -	10 12	Sekar et al. (2014) [54]
Seed	*Streptococcus pyogenes* *Bacillus subtilis* *Staphylococcus aureus* *Escherichia coli* *Pseudomonas aeruginosa*	15 - - - -	12 12 13 6.5 10	Bhat and Al-daihan (2014) [35]
Peel	*Escherichia coli* *Klebsiella pneumonia* *Proteus vulgarius* *Aspergillus fumigatus*	- - - -	14 13 15 15	Nethaji et al. (2015) [57]
Peel	*Methicillin-resistant S. aureus* (MRSA)	-	23.4	Rostinawati et al. (2018) [55]
Leaf	*Pseudomonas aeruginosa* multi-resistant	-	20.53	Sulistiyaningsih et al. (2018) [56]
Peel	*Staphylococcus aureus* *Listeria monocytogenes* *Escherichia coli* *Vibrio campbellii* *Vibrio parahaemolyticus Vibrio anguillarum* *Pseudomonas aeruginosa* *Salmonella enteritidis* *Candida albicans*	- - - - - - - -	- - - - - - - -	Phuong et al. (2020) [23]

**Table 6 molecules-26-07005-t006:** List of antioxidant assays performed on *N. ramboutan-ake*.

Part	Solvent	TPC	Antioxidant Assay	FRAP *	DPPH * (IC_50_)	Galvinoxyl (IC_50_)	Lipid Peroxidation (IC_50_)	H_2_O_2_ (IC_50_)	% Activity	References
Pulp	Methanol	433.78 ^a^	β-Carotene bleaching	-	-	-	-	-	76.58	Ikram et al. (2009) [69]
Leaf	Ethanol Aqueous	127.00 ^b^ 53.00 ^b^	ABTS * Galvinoxyl DPPH Lipid peroxidation	- -	0.24 ^f^ 3.70 ^f^	0.05 ^f^ 0.80 ^f^	0.30 ^f^ 3.13 ^f^	- -	- - - -	Ling et al. (2010) [68]
Pulp	Aqueous	199.07 ^c^	DPPH FRAP Metal chelating	73.06 ^d^	-	-	-	-	- - 72.78	Sulaiman and Ooi (2014) [24]
Peel	Methanol	306.04 ^b^	DPPH H_2_O_2_ scavenging FRAP Ferrous chelating Hydroxyl radical scavenging	-	0.019 ^f^	-	-	0.153 ^f^	- - - 14.48 14.28	Sukemi et al. (2015) [66]
Peel	n- hexane Ethyl acetate Methanol	-	DPPH	-	>1000 ^g^ 468.24 ^g^ 57.389 ^g^	-	-	-	36.74 82.92 95.77	Fadhli et al. (2018) [65]
Peel	Ethanol Aqueous	1.00 ^b^ 1.07 ^b^	DPPH FRAP	0.476 ^e^ 0.508 ^e^	-	-	-	-	88.90 80.40	Sopee et al. (2019) [67]

^a^ In mg of gallic acid per hundred g; ^b^ in mg of gallic acid per g; ^c^ in µg of gallic acid per g; ^d^ in µg of trolox per g; ^e^ in mg of ferrous (II) ion per g; ^f^ in mg per mL; ^g^ in µg per mL. * Abbreviations: ABTS: 2,2′-Azino-bis(3-ethylbenzothiazoline-6-sulfonic acid) diammonium salt; DPPH: 2,2-diphenyl-1-picrylhydrazyl; FRAP: ferric reducing antioxidant power.

**Table 7 molecules-26-07005-t007:** A comparison summary of studies conducted on *N. lappaceum* and *N. ramboutan-ake*.

Aspects	*N. lappaceum*				*N. ramboutan-ake*			
	Seed	Pulp	Leaf	Rind/ Peel	Seed	Pulp	Leaf	Rind/ Peel
Antioxidants	**√**	**√**	**√**	**√**		**√**	**√**	**√**
Anti-neoplastic	**√**			**√**				**√**
Anti-microbial	**√**		**√**	**√**	**√**			**√**
Hypoglycemic				**√**	**√**			
Anti-aging	**√**	**√**		**√**				

## Data Availability

Not applicable.

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
