# Peer review of "Review of Nephelium lappaceum and Nephelium ramboutan-ake: A High Potential Supplement"

_molecules, 2021, doi:10.3390/molecules26227005_

Round 1

Reviewer 1 Report

The paper entitled ‘Review on of Nephelium Lappaceum and Nephelium Ramboutan-Ake: A high potential supplement’ aimed to biochemical and biological activities characterization of the two plants. Authors decided to focus on their chemical profile, antioxidant, anticancer and antimicrobial activities. In my opinion, the paper is well organized and readable. References are adequate. In my opinion the subject of the paper is suitable to Molecules nevertheless the paper should be enriched in additional information:

  1. Authors present ‘other biological activities’ very briefly. I know that the plants are not detailed characterized nevertheless authors should try to find additional information (including mechanisms of action) about the aforementioned biological activities of the plants.
  2. I think that additional subsection (in form of table?) that will compare activity and use of the two plants will significantly improve the quality of the paper.
  3. References: position 74. Should be removed

Reviewer 2 Report

The manuscript presented by Jia Ling Tsong and cols, Nephelium Lappaceum and Nephelium Rambou- tan-Ake: A high potential supplement. It is an exciting topic since recently fruits and plants have been considered coadjuvants to treat health disorders due to theirs nutraceutical value.

The manuscript is well written and readable; however, a few typo errors should be corrected, as in line 101, among others. 

Some paragraphs are too long and contain many details; please reduce the size of manuscript length (especially in sections with few cites, summarize the results and experiments instead of a long narrative).

Please include the proper citation in line 61

Please reduce table 3 since there are carbohydrates and organic acids and just two references.

Tables 4 and 6 include units in the table and reduce footnotes in order to improve comprehension. Include a description of all abbreviations FRAP etc.

In section 4, Other Biological Activities, please include both species in each activity; the text is evident for each one of them. (No need to have two sections with the same effects). Also due to there are few manuscripts described there.

Please change the words “anti-diabetic” for hypoglycemic actions, exact with anti-neoplastic effects instead of anti-cancer.

In the conclusion second paragraph, the words “studies and lack” are too repetitive; please rephrase the sentence

Reviewer 3 Report

This manuscript detailed introduction to exploring the functions of Nephelium lappaceum (N. lappaceum) and Nephelium ramboutan-ake (N. ramboutan-ake), which highlighted the antioxidant, biological activities (anti-cancer, anti-microbial, anti-diabetic and anti-aging), and chemical contents from different parts of N. lappaceum and N. ramboutan-ake. Antioxidant activities and capabilities varies in each part of the fruit. Phenolic and flavonoid compounds are responsible for the antioxidant activities and other biological activities. In vitro and in vivo studies indicated that N. lappaceum carries pharmaceutical potential such as alleviating oxidative-stress related diseases. Such as red and yellow peel of N. lappaceum exhibit anti-proliferative effect on breast cancer cell and osteosarcoma cancer cell. The methanol peel extract successfully reduced cancer cell viability via inducing HepG2 apoptosis. In addition, the anti-diabetic effect of N. lappaceum was further studied in vivo. There was a reduction of α-glucosidase and α-amylase enzyme activities. Moreover, rat treated with N. lappaceum extract shows signs of anti-hyperglycemia such as reduction in blood glucose level and improvement of glucose tolerance. However, the lack of studies of N. ramboutan-ake especially human studies are lacking which resulted the underutilized of this fruit. Thus, the bioactive components responsible for the biological activities still remained unknown. this review has highlighted the potential and importance of N. lappaceum and N. ramboutan-ake. This manuscript content is suitable for publication in Molecules.

Round 2

Reviewer 1 Report

Authors improved the manuscript in accordance with reviewer suggestions. The current form of paper is more informative than in previous version. In my opinion, the paper can be accepted to publication.

Author Response

This manuscript is a resubmission of an earlier submission. The following is a list of the peer review reports and author responses from that submission.

Round 1

Reviewer 1 Report

Here are a few of my critical remarks on the manuscript: 
1. The title of the manuscript is incorrect and does not encourage you to read the entire potential article. 
2. The manuscript requires severe corrections, e.g., line 52 - tannins are not flavonoids. There are many such mistakes. 
3. In Table 2, we either give chemical or common names. What is astragalin? 
4. The authors did not include all the items of literature, e.g. Jahurul et al. - see - https://doi.org/10.1016/j.tifs.2020.03.016 and many others. 
5. Moreover, the literature is entirely unformatted.
6. There are no chemical structures or charts which would greatly assist in reading the manuscript.
7. It would be worth describing in detail and referring to the molecular mechanisms of action of the described species. This review would be interesting if the authors would refer to the published facts rather than simply putting the data together.

Reviewer 2 Report

The review article by Tsong Jia Ling et al. aims at highlighting the chemical profile, antioxidant, and biological activities of N. lappaceum and N. ramboutan-ake.

  • These are tropical fruits with some in vitro findings on their potential to counterbalance oxidative stress and resulted conditions. Human studies are lacking.
  • The present article seems to only sum the existing references on the above in vitro activities of different parts of the fruits.
  • No comparison or critique and assessment of the studies has been included.
  • No identification of any methodological assays has been conducted.
  • An extensive English language editing is required.